# Salivary Protein Profile in Patients with Recurrent Aphthous Stomatitis: A Pilot Proteomic Study

**DOI:** 10.3390/ijms26167878

**Published:** 2025-08-15

**Authors:** Francesco Franco, Nima Namarvari, Alessio Gambino, Federica Romano, Barbara Pergolizzi, Jianjian Zhang, Giuliana Abbadessa, Barbara Mognetti, Adriano Ceccarelli, Paolo Giacomo Arduino, Giovanni Nicolao Berta

**Affiliations:** 1Department of Clinical and Biological Sciences, University of Turin, 10043 Orbassano, Italy; francesco.franco@unito.it (F.F.); nima.namarvari@unito.it (N.N.); barbara.pergolizzi@unito.it (B.P.); jianjian.zhang@unito.it (J.Z.); giuliana.abbadessa@unito.it (G.A.); adriano.ceccarelli@unito.it (A.C.); 2Department of Surgical Sciences, CIR Dental School, University of Turin, 10126 Turin, Italy; alessio.gambino@unito.it (A.G.); federica.romano@unito.it (F.R.); paologiacomo.arduino@unito.it (P.G.A.); 3Department of Life Sciences and Systems Biology, University of Turin, Via Accademia Albertina 13, 10123 Turin, Italy; barbara.mognetti@unito.it; 4Neurosciences Institute of the Cavalieri-Ottolenghi Foundation, 10043 Orbassano, Italy

**Keywords:** cystatin, recurrent aphthous stomatitis, proteomic

## Abstract

Recurrent aphthous stomatitis (RAS) is the most common ulcerative disorder of the oral cavity, although its etiology is still unknown. The present study aimed to identify the proteomic profile associated with the RAS inflammatory process, thereby enhancing our understanding of its etiopathogenesis. We compared salivary protein profiles of RAS patients during an active episode of oral ulceration (30 patients, mean age 36.9) to those from healthy donors without a history of RAS (30 healthy subjects, mean age 37.9). Using 2D-electrophoresis and mass spectrometry (MALDI-TOF) analysis, we identified 17 proteins that were differentially expressed in the two groups. Notably, Cystatin SN (CST1) appeared to be significantly downregulated in RAS patients. These findings were validated by Western blot analysis: CST1 was detected in only 3 of the 30 RAS cases, while it was strongly expressed in all the healthy subjects. Although preliminary, our results suggest a potential role for CST1 in the etiopathogenesis of RAS. Interestingly, the relative absence of CST1 in RAS patients seems to align with some clinical and molecular features of this disease.

## 1. Introduction

Recurrent aphthous stomatitis (RAS) is an oral ulcerative disorder, characterized by recurrent episodes of mouth ulcers in otherwise healthy individuals [1]. The word “aphthous” derives from the ancient Greek verb “aphthein”, a term first used by Hippocrates in 400 BC to indicate a burning sensation of the airways or to describe painful ulcers of the oral cavity [2]. RAS is widely considered one of the most common oral diseases, ranging from a frequency of 5–25% up to 5–60% among the general population. It shows no significant sex predilection and typically affects adolescents and young adults between 10 and 40 years of age [3,4]. Three clinical variants of RAS are commonly recognized—minor, major, and herpetiform—which differ in the number and size of lesions, their location, and the duration of the outbreaks [5]. Its etiology is still unknown, although it is considered a multifactorial disorder with several pathogenetic routes and potential trigger factors (which can occur alone or in association with others), such as genetic predisposition, deficiency of microelements, systemic disorders, viral and bacterial infections, food allergies, hormonal factors, mechanical injuries, and anxiety [4,5,6,7,8]. In the last decade, its complex and poorly understood pathogenesis has been increasingly investigated using advanced biomolecular techniques aimed at identifying novel biomarkers. Among these, the proteomic analysis stands out as one of the most promising approaches, enabling the characterization of proteins and their post-translational modifications across many tissues and body fluids, including saliva samples [9]. The relatively simple and non-invasive method to collect salivary samples enhances the value of salivary proteomics in the identification of new molecular markers for both systemic and oral disorders [10,11,12]. So far, salivary biomarkers have been investigated across a vast spectrum of oral disorders. In autoimmune mucocutaneous diseases, such as Pemphigus Vulgaris and mucous membrane pemphigoid, differential expression profiles of desmoglein 1–3 and BP180 have been observed between cases and healthy controls [13,14]. In oral lichen planus, altered salivary cortisol levels have been reported in affected individuals compared to controls [15,16,17]. Furthermore, the analysis of the salivary proteome in patients with oral premalignant disorders has revealed an altered expression of salivary CK10 in those affected by oral leukoplakia (OL) compared to controls [18], along with a differential expression of complement component 4, endothelin 1, and lactate dehydrogenase between OL and oral squamous cell carcinoma (OSCC) [19,20]. Increasing evidence also supports the diagnostic potential of multiple salivary biomarkers in OSCC, especially interleukins (IL)-8 and IL1-β [21], and of Ki-67 and Cyclin D1 [22].

Concerning RAS, current evidence from salivary proteomics is still preliminary, being either divergent or, in some cases, conflicting [7,8]. Several reports on salivary proteomics in RAS patients have focused on the detection of specific protein/peptide targets, mostly related to the role of stress/anxiety in the onset of RAS, or immunogenic/inflammatory pathways [23,24,25,26,27,28,29,30,31,32,33,34]. The role of stress and anxiety was inferred from the quantification of salivary levels of cortisol. While some reports showed significantly higher salivary cortisol levels in RAS patients when compared to healthy individuals [23,24,25], others found overlapping values between the two groups [26,27]. Adding further complexity, the evaluation of oxidative stress-related salivary enzymes yielded conflicting results: some studies reported no significant differences in total antioxidant capacity and total oxidant status between RAS patients and healthy controls [28,29], while others detected increased levels of antioxidant enzymes, such as superoxide dismutase, catalase, and glutathione peroxidase in RAS patients [30,31]. Moreover, the analysis of the inflammatory/immune triggers of RAS through salivary proteomics identified tumor necrosis factor (TNF)-α as a key mediator, with significantly higher salivary levels observed both during acute RAS episodes [32,33,34] and remission phases [32], compared to healthy counterparts.

Therefore, the present study aimed at identifying a specific proteomic profile associated with the RAS inflammatory process, giving further insights into its etiopathogenesis. To this end, we compared salivary protein profiles of RAS patients during an active episode of oral ulceration with those from healthy subjects without a history of RAS.

## 2. Results

### 2.1. Patients

Based on the inclusion and exclusion criteria described in the Materials and Methods section, we consecutively recruited 60 participants. Of these, 30 were healthy individuals with no lifetime history of RAS (H group, 15 females and 15 males, mean age 34.2 ± 7.8 years), and 30 were RAS patients (R group, 14 females and 16 males, mean age 36.9 ± 8.6 years). The two groups were matched for age and gender. Among RAS patients, 22 (73.3%) had the minor subtype and 8 (26.7%) the major subtype of the disease, while no cases of herpetiform RAS were observed.

Notably, nearly half of the healthy subjects reported having experienced one or two isolated episodes of oral aphthae in their lifetime, typically self-limiting and associated with local triggers such as trauma, stress, or dietary irritants.

### 2.2. Salivary Proteomic Signature

Figure 1 shows the two-dimensional gel electrophoresis (2-DE) analysis performed on salivary proteins from a subset of clinically relevant subjects within the enrolled cohorts.

Comparison between the two groups showed a higher number of protein spots in the healthy samples compared to those from RAS patients (118 in the healthy group vs. 75 in the RAS group), especially within the range of 150 kDa to 8 kDa.

Eighty-five spots from healthy subjects and 73 from RAS patients were manually excised from the gels and subjected to MALDI-TOF-MS analysis to identify the corresponding proteins in fingerprinting mode. The spots unambiguously identified via MALDI-TOF-MS are numbered and circled in red in Figure 1. Seventeen proteins were identified: expectation value, molecular weight, protein ID, and functional categories are reported in Table 1 (healthy subjects) and Table 2 (RAS patients).

Several spots corresponded to either different isoforms of the same protein, or to the same protein with different migration patterns, likely reflecting post-translational modifications. Functional category analysis revealed that the predominant proteins were associated both with the immune system and hydroxylases/anti-hydrolases activities.

Among the 17 identified proteins, a recurrent spot located in the lowest section of the SDS-PAGE (≅17 kDa) was particularly noteworthy. This spot (see Figure 1, spots 9) showed a markedly higher intensity in the healthy group and was less expressed in RAS patients. MALDI-TOF-MS analysis identified this spot as Cystatin SN (Cystatin 1, CST1).

### 2.3. Validation of Selected Protein by Western Blot Analysis

Based on the proteomic results highlighting CST1 as the most significantly differentially expressed protein, we focused our subsequent analysis on the validation of its distinct expression between the healthy and RAS groups. Western blot (WB) analysis was performed using the total protein extracted from the saliva samples. As expected, a band of 16.6 kDa was detected, consistent with the molecular weight of CST1. A marked difference in CST1 expression was observed: CST1 was undetectable in the majority of RAS cases, whereas strong CST1 expression was consistently observed in all healthy samples. Figure 2 shows a representative WB illustrating CST1 expression across the two experimental groups.

The results of the WB analysis conducted on the 60 salivary samples included in the study are listed in Table 3. Briefly, CST1 was robustly expressed in all healthy subjects while in the majority of samples from RAS patients, its expression was absent or barely detectable, with only 3 out of 30 (10%) patients showing CST1 expression levels (Appendix A). The difference between the two groups was statistically significant (*p* < 0.001). Notably, a comparable proportion of CST1 expression was observed in patients with major and minor RAS (1 out of 8 vs. 2 out of 22, respectively), suggesting no subtype-specific difference.

## 3. Discussion

The identification of reliable molecular markers that enhance our knowledge about the etiopathogenesis or clinical progression of a disease, whether as indicators of disease exacerbation or as potential therapeutic targets of new drugs, is a central goal in both basic and clinical research. This endeavor lies at the core of the emerging paradigm of personalized medicine [35].

In this regard, proteomics offers a powerful tool to investigate protein expression in various biological samples, including biological fluids and tissue specimens. Among these, the salivary secretome has garnered growing interest, as it reflects many systemic and local pathologies [9].

Despite the high prevalence of RAS, its etiopathogenesis is still poorly understood. While it typically shows a benign clinical progression, and lesions tend to resolve spontaneously without scarring, its recurrence remains extremely variable and might significantly impair patients’ quality of life. In the most severe cases, as in the major RAS forms, patients report intense burning and itching sensations of the oral mucosa that can last for days or weeks, necessitating frequent use of topical and/or systemic corticosteroids for immediate relief [36].

In the present study, we attempted to characterize the salivary proteomic profile of patients with active RAS and to compare it with age- and sex-matched healthy individuals. Using MALDI-TOF mass spectrometry, we identified significant differences in the protein expression between the two groups. Among the 17 identified proteins, immunoglobulin kappa constant subtypes (IGKC, IGKCγ1 IGKCα1) calgranulin A, fibrinogen beta-chain, and CST1 were differentially expressed and found predominantly in RAS patients. Although data on the role of salivary IGKC in oral disorders is lacking, its involvement could be linked to its role in antigen binding, antigen presentation, and immune response activation, all of which are relevant in RAS pathogenesis [6,7]. Conversely, calgranulin A is already recognized as a salivary biomarker in other chronic inflammatory oral disorders, such as periodontitis [37]. Concerning fibrinogen beta-chain, despite the absence of data on its role in oral medicine, its hyper-expression (indicative of a high rate of conversion of fibrinogen to fibrin) among RAS patients might be caused by the local ulcerative process, which leads to the onset of recurrent oral ulcers covered by a characteristic yellow-grey, fibrin-rich pseudomembrane. Among the proteins analysed, CST1 drew our attention due to marked differences in detection between RAS patients, irrespective of the RAS subtypes, and healthy subjects. CST1 is a saliva-secreted type II isoform of a protein belonging to the cystatin superfamily [38,39]. These proteins are widely expressed in mammals and other animal species even if phylogenetically very distant from humans (e.g., snakes, soft, and hard ticks [40,41,42]), and they are known for their potent cysteine protease inhibitory activity [43], although they participate in a wide range of biological functions [39,44,45]. The homeostasis of protease activity is finely regulated: an alteration of the balance between proteinases and their inhibitors has been implicated in various pathological processes, including cancers [46].

Notably, CST1 isoform is involved in the regulation of key cellular pathways, such as Wnt, GSK3, AKT [46], and IL-6 [47], linking it to inflammatory processes/immunological response [44,48,49,50], cell cycle regulation/cell senescence [51], antimicrobial activity [38,52], cancer progression, and metastatic spread [53,54]. Furthermore, it is a powerful inhibitor of proteases belonging to the papain family protein (e.g., some isoforms of cathepsins) whose deregulation has been shown to play a fundamental role in maintaining tissue integrity, as well as in tissue remodeling [45,55,56].

To the best of our knowledge, this is the first report suggesting a potential involvement of salivary CST1 in the pathophysiology of RAS, but a direct causal relationship could not be established due to the cross-sectional design of the study.

RAS lesions typically present as round or oval superficial ulcers [7]. Tissue integrity is a finely regulated process to which many factors can contribute. In this context, the proteinases contributing to the degradation of the extracellular matrix seem to play a pivotal role [57]. Indeed, our study revealed a marked reduction in salivary CST1 expression during the acute phase of RAS. CST1 is the main protease inhibitor among the cystatins secreted in saliva, and it is reported to maintain tissue integrity. We hypothesize that CST1 reduction can trigger a higher protease activity in oral tissues, but we did not directly assess protease activity or downstream inflammatory responses upon CST1 reduction. Based on the previous literature, CST1 expression has been linked to other oral disorders, such as periodontitis, where excessive proteolytic activation has been documented [55,58,59,60]. In this context, CST1 has recently been reported to be downregulated in patients affected by severe forms of periodontitis, pointing toward the hypothesis that CST1 may serve as a general biomarker of oral cavity health [61].

In the present study CST1 was consistently detected in healthy participants but was nearly absent in RAS cases (detected only in three cases). The underlying reason for the presence of CST1 in the saliva of three out of thirty RAS patients remains unclear. However, based on our experience, we speculate that occult blood contamination in saliva samples might have contributed to this finding due to the presence of CST1 in blood components. Indeed, microbleeding from inflamed oral mucosa can occur, highlighting the importance of a standardized and carefully controlled saliva collection protocol. Moreover, to better address this potential confounder, future studies should incorporate a rapid test for detecting blood contamination in saliva samples prior to initiating experimental procedures.

Despite the multifactorial nature of RAS pathogenesis, our findings align with the hypothesis that CST1 downregulation may play a role in this process. Specifically, protease inhibitors are essential in protecting host tissues from the proteolytic activity of bacteria and fungi. RAS exacerbation has been associated with the co-infection with various microorganisms [62,63], suggesting that CST1 deficiency might facilitate prolonged microbial-induced destruction of the oral mucosa during the RAS episodes [44,52]. Overall, these results open new opportunities for research aimed at functionally characterizing the role of CST1 in the oral environment and elucidating the biological consequences of its downregulation, particularly in relation to protease activity, inflammatory responses, and mucosal integrity.

In the context of the deregulation of proteolytic activity, the role played by CST1 in antigen processing should also be considered, thereby influencing the quality of the immune response [44,50], which is central to RAS immunopathogenesis [64,65]. Some authors have proposed that cystatins regulate the immune system response, thus representing a new category of immunomodulators [49]. In addition, in vitro studies have shown that cystatins secreted in saliva can induce IL-6 production by oral fibroblasts via the NF-kB pathway [47,66]. Interleukin-6 is a pleomorphic cytokine that can play a crucial role in the immune response. Although it is commonly considered a pro-inflammatory molecule involved in many systemic disorders, it has been recently recognized as having important anti-inflammatory, anti-bacterial, and pro-resolutive effects, as well as in tissue regeneration. Its effects depend on the signaling pathway activated: classic-signaling (via the membrane-bound IL-6R) or trans-signaling (via soluble-receptor) [67]. Finally, although our findings suggest a potential involvement of CST1 in the pathophysiology of RAS, further studies are needed to determine whether CST1 contributes functionally to the disease process or simply reflects secondary alterations associated with the RAS condition.

This study has some limitations. First, the cross-sectional design limits our ability to draw conclusions regarding the causal or temporal relationship between CST1 expression and active RAS. Additionally, the relatively small sample size warrants caution, and our findings should be validated in larger cohorts that include different RAS subtypes, as well as patients in remission, to better characterize the dynamics of CST1 expression. RAS comprises clinically distinct subtypes, which may be associated with different underlying pathogenic mechanisms. Moreover, larger sample sizes would allow for the use of multivariable models to more effectively control for potential confounding factors.

A further concern is the relatively small number of proteins detected. As pointed out in a comparative review of salivary and plasma proteomes, the use of 2-DE and MALDI spectrometry allows for the detection of only a limited number of proteins, either from large-scale proteomics of the whole saliva or from parotid/submandibular/sublingual secretions, compared to more sensitive techniques, such as liquid chromatography-tandem mass spectrometry or capillary isoelectric focusing [68]. While 2-DE remains a useful method for protein separation and detection of isoforms, it is known to have limited sensitivity for low-abundance proteins and proteins with extreme physicochemical properties. Nevertheless, our findings are consistent with similar studies exploring the role of salivary biomarkers employing 2DE/MALDI spectrometry in a wide range of inflammatory, autoimmune, and premalignant oral disorders, such as Sjögren syndrome [69,70], periodontitis [71], and oral leukoplakia [18], which report the identification of 14 to 42 proteins from an original pool of 200–300 spots.

Despite these limitations, the present study demonstrated notable differences in the salivary proteomic profiles of RAS patients during acute manifestation of the disease compared to healthy subjects. Using MALDI-TOF-MS, we identified CST1 as a protein of significant biological relevance, whose relative absence in RAS patients may contribute to a better understanding of disease pathogenesis. This finding appears consistent with both clinical features and molecular mechanisms associated with RAS, positioning CST1 as a potential marker for further investigation.

## 4. Materials and Methods

### 4.1. Ethical Aspects and Patient Recruitment

This study was designed as a cross-sectional study with a case-control design and was performed according to the current standards of clinical research. The protocol was approved by the institutional Ethical Committee (n. 315/2016, “A.O.U. Città della Salute e della Scienza di Torino”), and all patients provided written informed consent in accordance with the Helsinki Declaration (1975, revised in 2013) prior to enrollment.

Patients in the acute phase of RAS and healthy donors were serially enrolled from the patients referred to the Section of Oral Medicine, CIR Dental School, University of Turin, between December 2020 and December 2023. Cases were recruited according to the following inclusion criteria: (i) chronic history of recurrent oral ulcers, defined as episodes occurring at least three times per year; (ii) presence of at least one oral round/oval, grey-white pseudomembranous ulcer, surrounded by an erythematous halo compatible with a clinical diagnosis of an aphthous lesion, identified as an exclusive oral sign by one of three oral physicians trained in oral medicine; (iii) no local or systemic therapy against oral ulcers during the previous 4 weeks; (iv) absence of history of autoimmune disease associated with onset of acute or chronic oral ulcers (Behçet disease, celiac disease, Crohn disease, erosive lichen planus, systemic lupus erythematosus, Wegener granulomatosis); (v) recent blood tests showing no evidence of hematinic deficiencies (iron, zinc, B2, B6, B12, and folic acid serum levels within range), anemic state (mean corpuscular volume, MCV, within 80–100 fL; mean corpuscular hemoglobin, MCH, within 26–32 pg), immunodeficiency or lymphocyte dysregulation (regular leukocyte formulae after complete blood count, regular proportion of T, B, NK lymphocytes after lymphocyte typing).

Current smokers, pregnant or breastfeeding females, patients under a systemic drug regimen for general health issues, and patients diagnosed as having plaque-related periodontal disease or inflammatory bowel disorders (fecal calprotectin < 50 μg/mg, no serum anti-tissue transglutaminase/anti-endomysium antibodies) were excluded from the study. Healthy donors were consecutively enrolled as age- and sex-matched according to the same inclusion criteria as the RAS group but without a history of RAS through their lives.

### 4.2. Salivary Collection and Protein Sample Preparation

Donors were asked to refrain from eating, drinking, or oral hygiene for at least one hour before collection (between 9 and 11 a.m.). Whole unstimulated saliva was collected from each participant according to the draining method [72]: subjects sat for 10 min in a noiseless room, with the head bent down and the mouth open, to allow the saliva to drip passively from the lower lip into a calibrated sterile tube. Total proteins from the salivary sol phase were precipitated overnight at −20 °C with an equal volume of acetone, methanol, and tributyl phosphate (TBP) (12:1:1 respectively) and centrifuged at 12,000× *g* in an Eppendorf microfuge for 20 min at 4 °C; the resulting pellet was then air-dried for 20 min at room temperature. Each pellet was dissolved in isoelectric focusing buffer containing 7.0 M urea, 2.0 M thiourea, 20 mM dithiotreitol (DTT) (Merck, Darmstadt, Germany), 4% CHAPS, 2% carrier ampholyte (pH 3–10), 30 mM TRIS (Amersham Biosciences, Amersham, UK), 1% phenylmethanesulfonyl fluoride (PMSF), 1% Triton X-100, 0.002% bromophenol blue (Sigma-Aldrich, St. Louis, MO, USA). Protein concentration was estimated using the Bradford protein assay (Bio-Rad Laboratories, Hercules, CA, USA) according to the manufacturer’s instructions.

### 4.3. Two-Dimensional Gel Electrophoresis

A 2-DE analysis was performed according to the manufacturer’s instructions (GE Healthcare, Milan, Italy) as previously reported [73]. Briefly, salivary samples containing 0.125 mg proteins were mixed with Destreak Rehydration solution (GE Healthcare, Milan, Italy) and 0.5% of IPG buffer (pH 3–10 NL) in a total volume of 140 μL rehydration buffer.

First dimension electrophoresis was performed in Immobiline DryStrips (7 cm, pH 3–10 NL; GE Healthcare, Milan, Italy) and passively rehydrated (20 V, 12 h). Isoelectric focusing with an IPGphor system (GE Healthcare, Milan, Italy) was performed at 200, 500, and 1000 V each for 1 h (step-N-hold), 1000–5000 V (gradient) for 30 min, and 5000 V (step-N-hold) for 3 h. The Immobiline Dry-Strips were then reduced for 30 min in SDS equilibration buffer (50 mM Tris-HCl pH 8.8, 6 M urea, 30% glycerol, 2% SDS) containing 1% (*w*/*v*) DTT, followed by 30 min alkylation in the same equilibration buffer with 2.5% (*w*/*v*) iodoacetamide instead of DTT.

After the first dimension, SDS-PAGE electrophoresis was run on gel precast criterion tgx (Bio-Rad Laboratories, Hercules, CA, USA), stained with colloidal Coomassie Blue G250 (Bio-Rad, Laboratories, Hercules, CA, USA), and scanned for an evaluation of the different profiles of protein distribution between cases and controls. A 2-DE analysis was performed on proteins collected from salivary samples of each enrolled subject, and at least two replicates from each single donor were performed.

### 4.4. Mass Spectrometry and Protein Identification

Protein spots were excised, transferred in 1.5 mL Eppendorf microtubes, and subjected to trypsin digestion (Proteomics grade, ROCHE, Milan, Italy). Afterward, 2 µL of each sample was mixed with an equal volume of α-cyano-4-hydroxycinnamic acid-based matrix (C8982 Sigma Life Science, St. Louis, MO, USA) saturated in 50% acetonitrile. Finally, 0.8 µL aliquots of this mixture were released on the metal target plate of a Microflex^®^ LRF MALDI-TOF mass spectrometer (Bruker Daltonics, Bremen, Germany) in reflector mode. The peptide mass fingerprint spectra were searched against the MASCOT (Matrix Science) and National Center for Biotechnology Information (NCBI, www.ncbi.nlm.nih.gov (accessed on 15 July 2025)) databases. The parameters used for the search of a protein database with PMF (peptide mass fingerprinting) were as follows: enzyme, trypsin; species, Homo sapiens; pI range, ±1; Mr range, ±20%; missed cleavage sites allowed, 1; minimum peptide hits, 4; mass tolerance, ±100 ppm; modifications, cysteine treated with iodoacetamide to carboxamidomethyl and methionine in the oxidized form.

### 4.5. Western Blotting to Validate Selected Protein

WB analysis was performed as previously reported [74]. Briefly, 30 µg of total proteins from each sample were loaded and run in a precast gel of acrylamide and transferred onto PVDF membranes (Bio-Rad Laboratories, Hercules, CA, USA). It was initially stained with ponceau red (Sigma Aldrich, MO, USA) (to confirm the presence of the same amount of proteins in all tested samples) and then probed with primary antibodies recognizing salivary cystatin SN/CST1 (Santa Cruz Biotechnology, Santa Cruz, CA, USA). An HRP-conjugated secondary antibody and ECL Western Blotting System (Bio-Rad Laboratories, Hercules, CA, USA) were used for detection. Immunoblots and Ponceau Red were digitally scanned using the ChemiDocTM Imaging System (Bio-Rad Laboratories, Hercules, CA, USA)

### 4.6. Statistical Analysis

Statistical analysis was performed using the SPSS statistical software package (version 28.0, Chicago, IL, USA). Data related to CST1 were summarized as absolute frequencies and percentages, and the difference between the case and control groups was assessed using Fisher’s exact test. The level of significance was set at 5%.

## Figures and Tables

**Figure 1 ijms-26-07878-f001:**
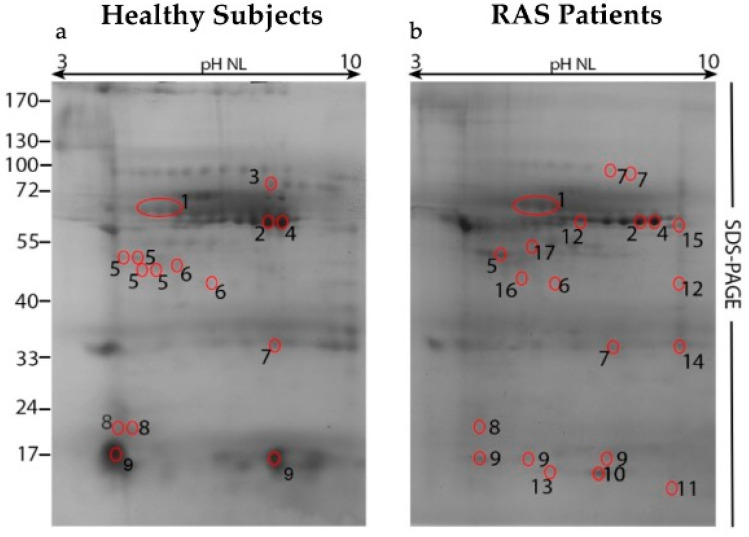
Two-dimensional gel electrophoresis (2-DE) of saliva sol phase samples obtained from (**a**) healthy (without a history of RAS) subjects and (**b**) RAS patients during an acute episode of ulceration. Unambiguously identified spots by MALDI-TOF-MS are circled in red and numbered. The spots are identified by averaging the data from all gels in each group.

**Figure 2 ijms-26-07878-f002:**
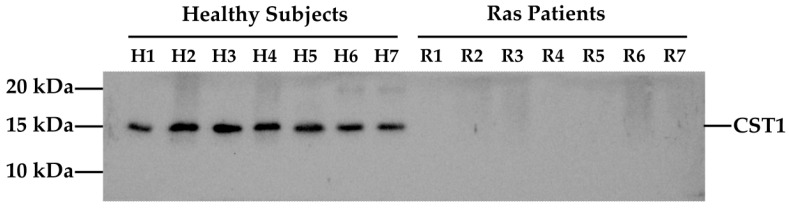
WB analysis of Cystatin SN (CST1) salivary samples from subjects involved in the study.

**Table 1 ijms-26-07878-t001:** Salivary proteins detected in healthy donors.

Functional Category	Spot Number	Protein	Protein ID *	Expect Value **	pI	MW
Inflammatory, antimicrobial	1	ALB	P02768	1.8 × 10^−5^	~5.0	71.3
	9	CST1	P01037	1.0 × 10^−7^	7.5	16.6
Hydrolases/Anti-hydrolases	2	AMY1A	P0DUB6	2.0 × 10^−14^	~6.5	58.4
	3	LTF	P02788	4.0 × 10^−5^	~8.0	80.14
	4	AMY2B	P19961	7.2 × 10^−5^	~6.7	42.021
Cytoskeletal	5	ACTB	P60709	0.00097	~5.3	42.052
	6	K1C10	P13645	0.0012	~5.13	58.827
Immune System	7	pIgR	P01833	0.0039	~7.0	84.4
Other Proteins	8	PIP	P12273	5.1 × 10^−10^	~5.0	16.572

ALB: albumin; CST1: Cystatin SN; AMY1A: α-amylase1A; AMY2B: α-amylase2B; LTF: lactotransferrin; ACTB: actin cytoplasmic I; K1C10: keratin I cytoplasm 10; pIgR: polymeric immunoglobulin receptor; PIP: prolactin-inducible protein. * Protein ID code from https://www.uniprot.org/ (accessed on 15 July 2025). ** Expected value for the peptide match. The number of times we would expect to obtain an equal or higher score, purely by chance. The lower this value, the more significant the result. (http://www.matrixscience.com/help/pmf_summaries_help.html) (accessed on 15 July 2025).

**Table 2 ijms-26-07878-t002:** Salivary proteins detected in RAS patients in acute phase.

Functional Category	Spot Number	Protein	Protein ID *	Expect Value **	pI	MW
Inflammatory, antimicrobial	1	ALB	P02768	8.1 × 10^−25^	~5.0	71.3
	9	CST1	P01037	1.0 × 10^−7^	~7.5	16.6
	10	CST4	P01036	0.00061	~7.5	16.4
	11	S100A8	P05109	0.0027	~5.0	10.8
Hydrolases/Anti-hydrolases	2	AMY1A	P0DUB6	5.1 × 10^−14^	~5.95	58.4
	2	AMY2B	P19961	2.0 × 10^−10^	~5.2	58.3
	12	AMY2A	P04746	4.0 × 10^−11^	~5.5	58.3
Cytoskeletal	5	ACTB	P60709	0.0001	~5.3	42.0
	6	K1C9	P35527	0.0021	~5.0	62.2
	13	K1C10	P13645	0.0016	~5.13	59.02
Immune System	14	IGKC	P01834	6.4 × 10^−6^	~5.5	11.9
	15	IGKCγ1	P01834	0.0023	~5.5	36.5
	16	IGKCα1	P01834	4.8 × 10^−5^	~5.5	38.4
	17	FGB	P02675	8.1 × 10^−7^	~5.5	56.5
	7	pIgR	P01833	6.9 × 10^−5^	~7.0	84.4
Other Proteins	8	PIP	P12273	2.5 × 10^−7^	~5.0	16.5

ALB: albumin; CST1: Cystatin SN; CST4: Cystatin S; S100A8: Calgranulin A; AMY1A: α-amylase1A; AMY2B: α-amylase2B; AMY2A: α-amylase ACTB: actin cytoplasmic I; K1C9: keratin I cytoplasm 9; K1C10: keratin I cytoplasm 10; IGKC: immunoglobulin kappa constant; IGKCγ1: immunoglobulin kappa constant gamma 1; IGKCα1: immunoglobulin kappa constant alpha 1; FBG: fibrinogen beta chain; pIgR: polymeric immunoglobulin receptor; PIP: prolactin-inducible protein. * Protein ID code from https://www.uniprot.org/ (accessed on 15 July 2025). ** Expected value for the peptide match. The number of times we would expect to obtain an equal or higher score, purely by chance. The lower this value, the more significant the result. (http://www.matrixscience.com/help/pmf_summaries_help.html) (accessed on 15 July 2025).

**Table 3 ijms-26-07878-t003:** CST1 expression in salivary samples collected from the two studied groups (+, strongly expressed; −, absent).

Healthy Subjects (H)	Level of CST1 Expression	RAS Patients (R)	Level of CST1 Expression
H1	+	R1	−
H2	+	R2	−
H3	+	R3	−
H4	+	R4	−
H5	+	R5	−
H6	+	R6	−
H7	+	R7	−
H8	+	R8	−
H9	+	R9	−
H10	+	R10	−
H11	+	R11	−
H12	+	R12	−
H13	+	R13	−
H14	+	R14	−
H15	+	R15	−
H16	+	R16	−
H17	+	R17	+
H18	+	R18	−
H19	+	R19	−
H20	+	R20	−
H21	+	R21	−
H22	+	R22	+
H23	+	R23	−
H24	+	R24	−
H25	+	R25	−
H26	+	R26	−
H27	+	R27	−
H28	+	R28	−
H29	+	R29	−
H30	+	R30	+

## Data Availability

The data presented in this study are available on request from the corresponding author.

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
