# Peer review of "Salivary Protein Profile in Patients with Recurrent Aphthous Stomatitis: A Pilot Proteomic Study"

_ijms, 2025, doi:10.3390/ijms26167878_

Round 1

Reviewer 1 Report

Comments and Suggestions for Authors

This paper studies the proteomic of saliva in patients with recurrent aphthous stomatitis (RAS), especially the difference in expression of Cystatin SN (CST1). RAS is a common oral ulcer disease with an unknown etiology. As a non-invasive method, saliva proteomic has the potential to find biomarkers, and the topic selection is of clinical significance and innovation, because there are differences in previous research results. This paper focuses on this and has certain research value. The manuscript can be accepted after a minor revision. The specific suggestions are as follows:

It is mentioned in the paper that "nearly half of the healthy subjects have reported occasional aphthous attacks", and such individuals may have potential immune or mucosal dysfunction, which contradicts the core definition of "no history of RAS" and may dilute the difference of CST1 expression between groups.

The paper speculates that "the CST1 positive in 3 patients with RAS may be due to trace blood pollution in saliva", but it does not explain how to verify this speculation.

RAS is divided into mild, severe and herpetic types, and its pathological mechanism may be different. However, the paper does not analyze it by subtype, which may cover up the difference of CST1 expression in different subtypes.

At present, the research only proves the correlation between CST1 and RAS, but not its direct role in the pathogenesis, and the logical chain is broken.

It is mentioned that CST1 is "the main inhibitor of salivary cysteine protease", but it has not been verified whether its deletion will enhance the protease activity or inflammatory reaction of oral mucosa.

The existing research can't distinguish between "the decrease of CST1 leads to RAS attack" and "the decrease of CST1 leads to RAS attack".

In this paper, only 2-DE+MALDI-TOF is used to identify differential proteins, while 2-DE has limited detection sensitivity for low-abundance proteins and may miss key molecules.

Fisher exact test was used to compare the differences of CST1 expression, but possible confounding factors (such as other variables besides age and sex) were not considered.

It is suggested that the following problems should be improved in future research. In terms of sample size, it is suggested to expand the sample size to include patients with RAS at different stages (active stage and remission stage) to analyze the dynamic changes of CST1 expression. In terms of detection methods, more sensitive techniques such as liquid chromatography-mass spectrometry (LC-MS/MS) can be used to improve the quantity and accuracy of protein identification. In terms of mechanism research, we can explore the effect of CST1 on protease activity and immune response through in vitro experiments (such as cell model), and clarify its mechanism in RAS. In addition, the potential confounding factors, such as stricter sample collection standards and avoiding blood pollution, are controlled to improve the reliability of the results.

Author Response

Reviewer 1

We would like to thank the Reviewer for their thoughtful review and we appreciate the reviewer’s valuable feedback. Your insightful comments helped us to strengthen the quality of our paper. The changes are tracked in the revised paper (in red).

Comment 1: It is mentioned in the paper that "nearly half of the healthy subjects have reported occasional aphthous attacks", and such individuals may have potential immune or mucosal dysfunction, which contradicts the core definition of "no history of RAS" and may dilute the difference of CST1 expression between groups.

Response 1: We thank the Reviewer for this observation. Although some control group participants reported sporadic oral stomatitis episodes, none had a clinical diagnosis of RAS or experienced recurrent episodes of aphthous stomatitis. We have now clarified this aspect in the manuscript. (lines 100-102 and line 308)

Comment 2: The paper speculates that "the CST1 positive in 3 patients with RAS may be due to trace blood pollution in saliva", but it does not explain how to verify this speculation.

Response 2: We acknowledge that our suggestion regarding trace blood contamination is speculative, although multiple studies have reported strong CST1 expression in blood. We have therefore revised the discussion section to clarify this point (Line 239-244).

Comment 3: RAS is divided into mild, severe and herpetic types, and its pathological mechanism may be different. However, the paper does not analyze it by subtype, which may cover up the difference of CST1 expression in different subtypes.

Response 3:  We thank the Reviewer for this observation. In our study cohort, 8 patients presented with the major form of RAS and 22 with the minor form. However, the limited sample size did not allow for a statistically robust stratification by RAS subtype. Notably, a comparable proportion of CST1 expression was observed in patients with major and minor RAS (1 out of 8 vs. 2 out of 22, respectively), suggesting no apparent subtype-specific difference within the current dataset.

We agree that RAS encompasses distinct clinical subtypes and that these may involve different underlying pathogenic mechanisms. We have now acknowledged this limitation in the revised manuscript and emphasized the need for larger studies to evaluate CST1 expression across different RAS subtypes in a statistically robust manner.

Comment 4: At present, the research only proves the correlation between CST1 and RAS, but not its direct role in the pathogenesis, and the logical chain is broken.

Response 4:  We appreciate the Reviewer’s insightful comment. We acknowledge that our current study demonstrates a correlation between decreased CST1 expression and active RAS, but does not establish a direct causal relationship. As discussed in the revised manuscript (lines 267-277), we have now clarified that our findings are observational in nature and represent an association rather than a mechanistic demonstration. We agree that future functional studies are necessary to elucidate the direct role of CST1 in the pathogenesis of RAS.

Comment 5: It is mentioned that CST1 is "the main inhibitor of salivary cysteine protease", but it has not been verified whether its deletion will enhance the protease activity or inflammatory reaction of oral mucosa.

Response 5: We thank the Reviewer for this request of clarification. We have now implemented the discussion section of the study in order to address this reviewer’s concern (line 250-254).

Comment 6: The existing research can't distinguish between "the decrease of CST1 leads to RAS attack" and "the decrease of CST1 leads to RAS attack".

Response 6: We appreciate the Reviewer’s comment to clarify the directionality of the observed association. As rightly pointed out, our cross-sectional design does not allow us to determine whether the reduction in CST1 is a cause or a consequence of RAS lesions. We have added this limitation explicitly in the revised Discussion section (lines 220-221).

Comment 7: In this paper, only 2-DE+MALDI-TOF is used to identify differential proteins, while 2-DE has limited detection sensitivity for low-abundance proteins and may miss key molecules.

Response 7: We thank the Reviewer for this request of clarification. We acknowledge in the manuscript (lines 278-285) the limitations of the employed proteomic approach identifying other techniques that could be applied to scale up this pilot research initiative. Despite some sensitivity limitations, the combined use of 2- DE and MALDI-TOF enabled us to screen the salivary proteome effectively, revealing a distinct expression pattern of CST1 between the experimental groups. To address the intrinsic limitations of this approach and validate our findings, Western blotting was subsequently performed on all salivary samples as a confirmatory step.

Comment 8: Fisher exact test was used to compare the differences of CST1 expression, but possible confounding factors (such as other variables besides age and sex) were not considered.

Response 8:  We thank the Reviewer for this comment. We acknowledge that our statistical analysis using Fisher’s exact test did not account for potential confounding variables beyond age and sex, which were addressed through the study design. Our primary objective was to conduct an initial exploratory assessment of CST1 expression differences between RAS patients and healthy controls. While we applied strict inclusion criteria, excluding individuals with systemic diseases, current medications, and smoking habits, we recognize that additional unmeasured factors such as dietary habits, alcohol consumption, or subclinical inflammation may influence CST1 expression. We have now acknowledged this limitation in the revised manuscript and emphasized the need for future studies incorporating multivariable models to account for potential confounders more comprehensively.

Reviewer 2 Report

Comments and Suggestions for Authors

The authors compared salivary proteins between recurrent aphthous stomatitis (RAS) patients and healthy donors using 2D-electrohpsresis coupled with MALDI-TOF. A total of 17 proteins were found to be differentially expressed. Among these, CST1 was significantly downregulated in the RAS group, a finding validated by western blotting. This study demonstrates a notable difference in protein expression between RAS and healthy groups in terms of proteomic profiling, although the protein identification coverage is very limited for a proteomics study. Specific concerns are outlined below:

  1. The separation performance of 2D-electrophoresis and MALDI-TOF is insufficient for salivary proteomics, where thousands of proteins can be detected. The analytical approach used here identified only a few differentially expressed proteins and is therefore not optimal for comprehensive proteomic analysis.
  2. While CST1 was found to be significantly downregulated in the RAS group, no experimental work was presented to investigate its biological function.
  3. Section 2.1 should be moved to the experimental section. And its wording may cause confusion, particularly in light of the statement in section 4.1: “However, despite the absence of a formal RAS diagnosis, nearly half of the healthy subjects reported having experienced occasional episodes of aphthous stomatitis during their lifetime.”
  4. The details regarding protein reduction, alkylation, and digestion are not clearly described.

Author Response

Reviewer 2:

We thank the Reviewer for the time spent in revising our paper and for the valuable comments. The changes are tracked in the revised paper (in red).

Comment 1: The separation performance of 2D-electrophoresis and MALDI-TOF is insufficient for salivary proteomics, where thousands of proteins can be detected. The analytical approach used here identified only a few differentially expressed proteins and is therefore not optimal for comprehensive proteomic analysis.

Response 1: We thank the Reviewer for this valuable observation. We acknowledged in the manuscript (lines 278-285) the limitations of the employed proteomic approach identifying other techniques that could be applied to scale up this pilot research initiative. Although the 2-DE/MALDI-TOF workflow had some sensitivity limitations, it allowed to screen the salivary proteome and revealed clear differences in CST1 expression between the study groups. To overcome these methodological limitations and confirm the results, all salivary samples were further analyzed by Western blotting.

Comment 2: While CST1 was found to be significantly downregulated in the RAS group, no experimental work was presented to investigate its biological function.

Response 2: We thank the Reviewer for this important note. We have revised and expanded the Discussion section to specifically address this concern (line 250-254).

Comment 3: Section 2.1 should be moved to the experimental section. And its wording may cause confusion, particularly in light of the statement in section 4.1: “However, despite the absence of a formal RAS diagnosis, nearly half of the healthy subjects reported having experienced occasional episodes of aphthous stomatitis during their lifetime.”

Response 3: We thank the Reviewer for the valuable comment. In response, we have revised the sentence for improved clarity and moved it to the experimental section, as suggested.

Comment 4: The details regarding protein reduction, alkylation, and digestion are not clearly described.

Response 4: We thank the Reviewer. We reported the details of protein reduction and alkylation inside paragraph 4.3 (lines 352-355). While the details regarding digestion are on paragraph 4.4 (line 364).

Round 2

Reviewer 2 Report

Comments and Suggestions for Authors

The authors have effectively addressed the previous concerns, and I recommend acceptance of the revised manuscript.